# Gray Matter Abnormalities in Patients with Complex Regional Pain Syndrome: A Systematic Review and Meta-Analysis of Voxel-Based Morphometry Studies

**DOI:** 10.3390/brainsci12081115

**Published:** 2022-08-22

**Authors:** Teng Ma, Ze-Yang Li, Ying Yu, Yang Yang, Min-Hua Ni, Hao Xie, Wen Wang, Yu-Xiang Huang, Jin-Lian Li, Guang-Bin Cui, Lin-Feng Yan

**Affiliations:** 1Functional and Molecular Imaging Key Lab of Shaanxi Province, Department of Radiology, Tangdu Hospital, Fourth Military Medical University, 569 Xinsi Road, Xi’an 710038, China; 2Student Brigade, Fourth Military Medical University, 169 Changle Road, Xi’an 710032, China

**Keywords:** complex regional pain syndrome, gray matter volume, voxel-based morphometry, magnetic resonance imaging, meta-analysis

## Abstract

Current findings on brain structural alterations in complex regional pain syndrome (CRPS) are heterogenous and controversial. This study aimed to perform a systematic review and meta-analysis to explore the significant gray matter volume (GMV) abnormalities between patients with CRPS and healthy controls (HCs). A systematic search of the PubMed, Web of Science, and MEDLINE databases was performed, updated through 27 January 2022. A total of five studies (93 CRPS patients and 106 HCs) were included. Peak coordinates and effect sizes were extracted and meta-analyzed by anisotropic effect size–signed differential mapping (AES-SDM). Heterogeneity, sensitivity, and publication bias of the main results were checked by the Q test, jackknife analysis, and the Egger test, respectively. Meta-regression analysis was performed to explore the potential impact of risk factors on GMV alterations in patients with CRPS. The main analysis exhibited that patients with CRPS had increased GMV in the left medial superior frontal gyrus (SFGmedial.L), left striatum, and an undefined area (2, 0, −8) that may be in hypothalamus, as well as decreased GMV in the corpus callosum (CC) (extending to right supplementary motor area (SMA.R), right median cingulate/paracingulate gyri (MCC.R)), and an undefined area (extending to the right caudate nucleus (CAU.R), and right thalamus (THA.R)). Meta-regression analysis showed a negative relationship between increased GMV in the SFGmedial.L and disease duration, and the percentage of female patients with CRPS. Brain structure abnormalities in the sensorimotor regions (e.g., SFGmedial.L, SMA.R, CAU.R, MCC.R, and THA.R) may be susceptible in patients with CRPS. Additionally, sex differences and disease duration may have a negative effect on the increased GMV in SFGmedial.L.

## 1. Introduction

Complex regional pain syndrome (CRPS), commonly caused by limb trauma, is a chronic pain that is characterized by spontaneous and evoked regional pain in the extremities [1,2]. The prevalence of CRPS is about 5.4~26.2 per 100,000 person-years with a female predominance [3,4]. One of the significant impairments in CRPS is pain-induced neuroplasticity in the central nervous system (CNS). For instance, it was reported that cognitive function impairments and abnormal body representation were showed in CRPS patients without brain injury [5]. Resting-state functional magnetic resonance imaging (Rs-fMRI) studies have shown that CRPS involves significant default-mode network (DMN) alterations by comparing CRPS and non-CRPS patients [6,7]. Although multiple factors are involved in the process of CRPS, insufficient understanding of the underlying mechanism leads to poor clinical intervention and management [8]. Thus, further study of CNS changes is of great significance for targeted treatment and improving intervention efficiency.

Neuroimaging studies have focused on the brain structure alterations involved in CRPS. Voxel-based morphometry (VBM) is one of the most important methods for measuring gray matter volume (GMV) abnormalities [9,10]. According to the accumulating VBM evidence, patients with CRPS show GMV abnormalities in specific brain regions (e.g., cingulate cortex and amygdala) that play a key role in somatosensory and emotional functions. [7]. In addition, a neuroimaging study reported that GM atrophy might be limited in the emotion-related brain regions including the insula, ventromedial prefrontal cortex (VMPFC), and nucleus accumbens (NAc) [11]. In the early stage of CRPS, patients showed decreased GMV in cerebral areas that are involved in spatial body perception, and somatosensory and limbic systems [12]. Although brain structure abnormalities have been found to be closely related to the dysfunctions within CRPS, the findings on GMV alterations in specific brain regions can be heterogenous and controversial. For instance, it was reported that GMV in the dorsomedial prefrontal cortex of patients with CRPS was higher than that of HCs [1], whereas Barad et al. found that patients with CRPS had higher GMV in the right hypothalamus, left dorsal putamen, and left inferior temporal lobe, as well as lower GMV in the left orbitofrontal cortex, left middle cingulate cortex, right middle cingulate cortex, left posterior middle cingulate cortex, left dorsal insula, and left anterior middle cingulate cortex at the same time [13]. Due to the limited research samples, heterogeneous and unreliable findings may exist in CRPS studies [14]. Therefore, it is of great necessity to perform a systematic review and meta-analysis exploring the reliable GMV alterations in CRPS.

In this study, we aimed to conduct a voxel-wise meta-analysis including only VBM studies to reduce potential bias and explore reliable GMV abnormalities between CRPS patients and HCs, providing a valuable reference for future research and clinical management.

## 2. Materials and Methods

### 2.1. Search Strategy and Study Selection

This systematic review and meta-analysis followed the PRISMA checklist (Appendix A) [15], and the protocol was registered in PROSPERO (registration number CRD42022307238) (https://www.crd.york.ac.uk/prospero/). A systematic search was performed using the PubMed, Web of Science, and MEDLINE databases, updated through 27 January 2022, with the items as follows: (“complex regional pain syndromes” OR “CRPS” OR “complex regional pain syndrome, type i” OR “complex regional pain syndrome type ii”) AND (“gray matter” OR “gray matter volume” OR “VBM” OR “voxel-based morphometry” OR “GMV” OR “GM”). The detailed search strategy is shown in Appendix A. Additionally, references from review articles were searched manually.

Studies meeting all the following criteria were included: (1) the exploration of GMV alterations between CRPS patients and HCs; (2) adult participants; (3) available coordinate information (x, y, z) and effect size (t value or z score) reported in the standard coordinate space, including Montreal Neurological Institute (MNI), or Talairach (TAL) coordinates; (4) VBM method used to measure the GMV alteration. Studies meeting one of the following criteria were excluded: (1) animal study; (2) VBM method not used; (3) not CRPS; (4) not original study; (5) children participants; (6) not related to GMV.

### 2.2. Quality Assessment and Data Extraction

A quality assessment of the studies included was performed using a 12-point checklist (Appendix A), which has commonly been used in neuroimaging meta-analysis studies [16,17]. Both demographic and technical information were collected and presented in this study. Available coordinate information (x, y, z) and effect size (t value or z score) of the GMV alteration between CRPS patients and HCs were extracted for meta-analysis, and the transformation between z score and t value was performed using the SDM website (https://www.sdmproject.com/utilities/?show=Statistics). All work was independently accomplished by two reviewers (TM and ZYL), and any controversial assessments were solved by the third reviewer (LFY).

### 2.3. Voxel-Wise Meta-Analysis

Significant GMV abnormality between CRPS patients and HCs was analyzed by using AES-SDM version 5.15 (www.sdmproject.com), which has been widely used in brain structure meta-analysis [18,19]. This process was reported in previous studies [20,21], and is briefly described as follows: The significant coordinates and effect sizes of GMV abnormality between CRPS patients and HCs for each dataset were extracted and recreated as a new statistical map in the MNI space by using an anisotropic Gaussian kernel. Then, the voxel-wise calculation of the random-effects mean of the dataset maps, weighted by the sample size, intra-dataset variability, and between-dataset heterogeneity, was used to generate a mean map. The SDM default thresholds (FWHM = 20 mm, *p* = 0.005, peak height Z  =  1, and cluster extent  =  10 voxels) were used in this study. The 50 randomization test was used to generate stable results [22]. No subgroup analysis was performed because of limited datasets.

### 2.4. Heterogeneity, Sensitivity, and Publication Bias Assessment

The default Q maps of SDM were used to check the specific brain regions that present between-study heterogeneity when compared with a global set of voxels. The *p* threshold, peak height Z, and cluster extent in the random-effects model were set to 0.005, 1, and 10 voxels, respectively [21]. The robustness of the meta-analysis results was assessed by a whole-brain, voxel-based jackknife sensitivity analysis, which was performed by repeating the same steps as the main analysis after removing each study, one by one. Additionally, the total repeated times of the main results in all or most analysis results were regarded as reliable. Potential publication bias was checked by the Egger test with an SDM default. A *p*-value less than 0.05 was regarded as having publication bias.

### 2.5. Meta-Regression Analysis

The linear model in SDM was selected to perform meta-regression. Conservative thresholds (probability = 0.0005; peak height threshold = 1.000; extent threshold = 10) were used to explore the relationship between risk factors (age, sexual difference, pain score (VAS), and disease duration) and GMV alterations in CRPS.

## 3. Results

### 3.1. Characteristics of Studies Included

A total of 411 studies were found from three databases; 129 results were removed because of duplication, 270 of the remaining 282 studies were removed after screening the title and abstract, and 7 of the remaining 12 studies were removed after browsing the full text, meaning 5 studies were included in this meta-analysis [1,11,12,13,14]. A detailed search flow diagram is shown in Figure 1. No study was found from searching the references. In the end, a total of 199 participants (93 CRPS patients and 106 HCs, 50 male and 149 female) aged between 40.5 and 54.42 years old were included in this study. Demographic information, including author, publication year, sample size, age, disease duration, tools for pain severity assessment, and quality assessment of the study is shown in Table 1. Technical information, including the main GMV abnormality findings in CRPS patients, magnetic field, neuroimaging method, standard space, voxel size, and *p*-value, is shown in Table 2.

### 3.2. Main Analysis of GMV Abnormality between CRPS Patients and HCs

Compared with HCs, patients with CRPS showed significantly and consistently increased GMV in the medial left superior frontal gyrus (SFGmedial) (MNI: 2, 52, 20; SDM-Z = 1.005; *p* < 0.001; voxels = 324; jackknife analysis: 3/5), left striatum (MNI: −22, −4, −6; SDM-Z = 1.029; *p* < 0.001; voxels = 79; jackknife analysis: 4/5), and an undefined area (MNI: 2, 0, −8; SDM-Z = 1.034; *p* < 0.001; voxels = 56; jackknife analysis: 3/5) (Table 3; Figure 2), whereas decreased GMV in the CC (MNI: −2, −18, 26; SDM-Z = −1.769; *p* < 0.001; voxels = 1601; jackknife analysis: 4/5) and an undefined area (MNI: −8, −18, 20; SDM-Z = −1.693; *p* < 0.001; voxels = 312; jackknife analysis: 3/5) (Table 3; Figure 2) was observed. Detailed jackknife analysis information is shown in Table 4.

### 3.3. Heterogeneity and Publication Bias Assessment

No heterogeneity was found between specific brain regions and the global set of voxels (Q positive = 0.299, *p* > 0.05; Q negative = 0.951, *p* > 0.05) (Appendix A, Figure 3). The Egger test showed no publication bias in the SFGmedial (bias: −0.30, t: −0.13, df: 3, *p* > 0.05), left striatum (bias: 1.50, t: 1.17, df: 3, *p* > 0.05), and CC (bias: −0.39, t: −1.45, df: 3, *p* > 0.05), as well as in two undefined areas ((2, 0, −8) (bias: 7.41, t: 1.93, df: 3, *p* > 0.05), (−8, −18, 20) (bias: 1.36, t: 0.32, df: 3, *p* > 0.05)). Funnel plots are shown in Figure 4.

### 3.4. Meta-Regression Analysis

By performing a meta-regression analysis, the disease duration (month) (r = 0.5089, *p* < 0.0005) and the percentage of female patients (r = 0.9602, *p* < 0.0005) showed a negative relationship with the increased GMV of the SFGmedial.L in CRPS patients (Figure 5), whereas the age and pain score (VAS) showed no significant relationship with CRPS.

## 4. Discussion

In this study, a systematic review and meta-analysis of GMV abnormalities was first performed within CRPS patients. The main analysis indicated that increased GMV in the SFGmedial.L, left striatum, and an undefined area (2, 0, −8) was shown in CRPS patients, whereas decreased GMV was found in the CC area (extending to the SMA.R and MCC.R) and an undefined area (extending to the CAU.R and THA.R). In addition, the meta-regression analysis suggested that disease duration and the percentage of female patients might have a negative effect on the increased GMV in the SFGmedial.L in CRPS patients.

Interestingly, our meta-analysis results found higher GMV in the SFGmedial.L in CRPS patients than HCs, which demonstrated a negative relationship with the disease duration and percentage of female patients. Previous systematic review and meta-analysis studies reported significant gray matter (GM) deficits in the SFG in several neuropsychological disorders [23,24]. It is well known that the SFG in humans plays a crucial role in cognitive function, especially in the working memory (WM) [25,26]. The SFGmedial is associated with various neuropsychological disorders, including schizophrenia (SCZ), obsessive compulsive disorder (OCD), attention deficit hyperactivity disorder (ADHD), and bipolar disorder [27,28,29]. Combined with the evidence that long-term chronic pain stimulation can also induce a series of emotional disorders (e.g., anxiety, depression) [30,31] and the female predominance in CRPS, we speculated that the increased GMV in the SFGmedial may be a functional compensation in patients with CRPS, which may be attenuated by the disease duration and female ratio. Our meta-analysis also showed a larger GMV in the left striatum, which was regarded as one of the most important nuclei in regulating motor function [32]. It has also been reported that the increased GMV of the left striatum is significantly involved in the process of essential tremor and functional movement disorders [33,34]. Taken together, we can hypothesize that the possible mechanism of persistent pain stimulation from the extremities impairs the motor function in patients with CRPS. According to the neuroimaging evidence, we found that the undefined area (2, 0, −8) may be in the hypothalamic area [35,36]. Interestingly, the clinical and neuroimaging studies showed the altered neurotransmitter system involved in the hypothalamus and functional connectivity alterations between the hypothalamus and distal nuclei may be an incentive for pain-related aversions (e.g., fatigue and emotional alterations), as well as a biomarker for migraine occurrence [37]. Therefore, the brain structure alteration within the hypothalamus may be a potential mechanism for CRPS-related clinical or neurophysiological symptoms.

In our study, the GMV of the SMA.R and THA.R was significantly decreased, which was consistent with the previous meta-analyses of neuropathic pain [38,39]. Krainik et al. identified the important role of the SMA in motor development in patients undergoing medial frontal lobectomy [40]. Motor skill impairment has also been found in healthy adults receiving SMA.R-guided transcranial magnetic stimulation (nTMS) [41]. Taken together, these findings suggest CRPS may induce the motor dysfunction of patients by impairing the brain structure in the SMA.R. Interestingly, patients with hypoactivation of the SMA showed the same motor dysfunction as medial frontal lobectomy patients, which suggested the possible functional compensation of greater GMV in the SFGmedial to withstand motor dysfunction [40]. Furthermore, our study also found decreased GMV in the THA.R, which may play a dual role in both sensory and motor dysfunction in CRPS patients. For instance, the THA was generally considered to be a very important relay of peripheral sensory information to the cortex [42]. It also plays a key role in generating and monitoring movement by establishing direct contact with the movement-related cortex (e.g., SMA) [43]; the motor THA has also been an important target in treating tremors [44]. In addition, GM atrophy in the CAU was consistent with a previous neuroimaging study in the elderly with slower walking speeds [45]. The clinical and animal studies also identified the crucial role of the CAU in both posture and motor function maintenance [46]. Anatomical evidence revealed CAU lesions were significantly involved in motor apraxia [47]. The CRPS patients also showed a lower GMV in the MCC.R in our study, which is involved in the process of first-episode schizophrenia [48]. Interestingly, previous systematic reviews and meta-analyses of chronic migraine and fibromyalgia showed significantly decreased GMV in the bilateral anterior cingulate/paracingulate cortex (ACC) [49,50], which is involved in bipolar disorder [51]. Therefore, we may hypothesize that different types of chronic pain can impair the different subregions of the cingulate/paracingulate cortex to induce various psychological disorders.

This meta-regression analysis showed the female percentage of CRPS patients might negatively affect the increased GMV in the SFGmedial, which was consistent with the fact that female patients were more prone to CRPS (female:male = 4:1) [52]. Additionally, the disease duration negatively correlated with an increased GMV in the SFGmedial, suggesting long-term pain stimulation might induce worse impairments in the brain structure. Taken together, the potential effects of sex differences and disease duration should be considered in future CRPS studies. Furthermore, it was suggested to explore other risks (e.g., age and pain severity score) of the disease to further expand these datasets. It must also be pointed out that the sociodemographic factors, prior interventions (e.g., medication), prior behavioral health history/comorbidities (prevalence of anxiety/depression), and other potential confounding variables that were unavailable in the included studies might also have an effect on the brain structural alterations in patients with CRPS.

## 5. Limitations

This study had some limitations, which are declared as follows: Firstly, the study only extracted the available peak coordinate information, which is a common drawback in neuroimaging meta-analyses. Secondly, only five studies from three databases were included and, because of this, a subgroup analysis of CRPS (type I and type II) could not be performed. Thirdly, the meta-regression results should be treated cautiously due to the small size of the datasets, as well as the risk factors that were suggested for comprehensive exploration with larger datasets. Fourthly, the reliability of the heterogeneity, sensitivity, and publication bias assessment may be limited, which can be enhanced by including more studies in the future. Fifthly, the lesser studies included attenuated the reliability of the main findings in this study.

## 6. Conclusions

In conclusion, patients with CRPS may be susceptible to GMV impairments in sensorimotor cerebral areas. The increased GMV of the SFGmedial may carry out a functional compensation due to motor function deficits. In the process of CRPS, the SFGmedial and THA may play dual roles in motor and neuropsychological disorders, whereas the SMA, CAU, and MCC affect motor and emotional dysfunction. In short, our study provided a reference for a better understanding and exploration of CRPS.

## Figures and Tables

**Figure 1 brainsci-12-01115-f001:**
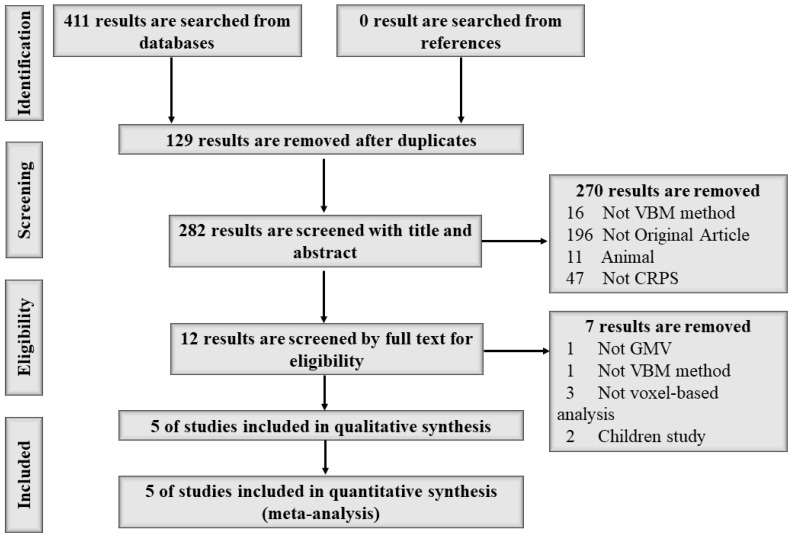
The search flow diagram following PRISMA guidelines.

**Figure 2 brainsci-12-01115-f002:**
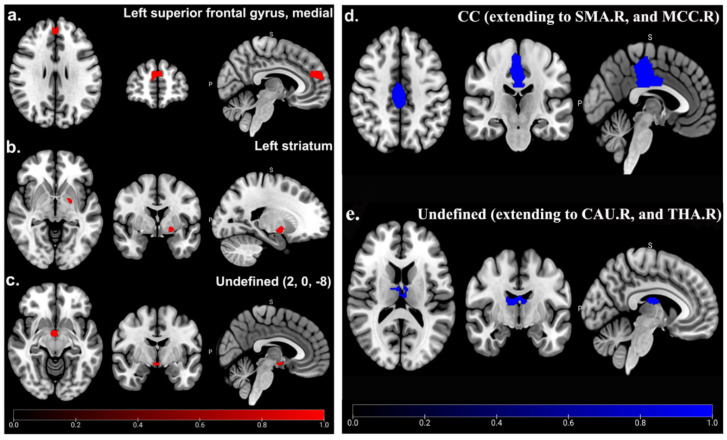
GMV abnormality between CRPS patients and HCs in main analysis. Significant increased (red) GMV of SFGmedial.L (**a**), left striatum (**b**), and undefined area (2, 0, −8) (**c**), as well as decreased (blue) GMV of CC (extending to SMA.R and MCC.R) (**d**), and undefined area (extending to CAU.R and THA.R) (**e**) in CRPS patients compared to HCs.

**Figure 3 brainsci-12-01115-f003:**
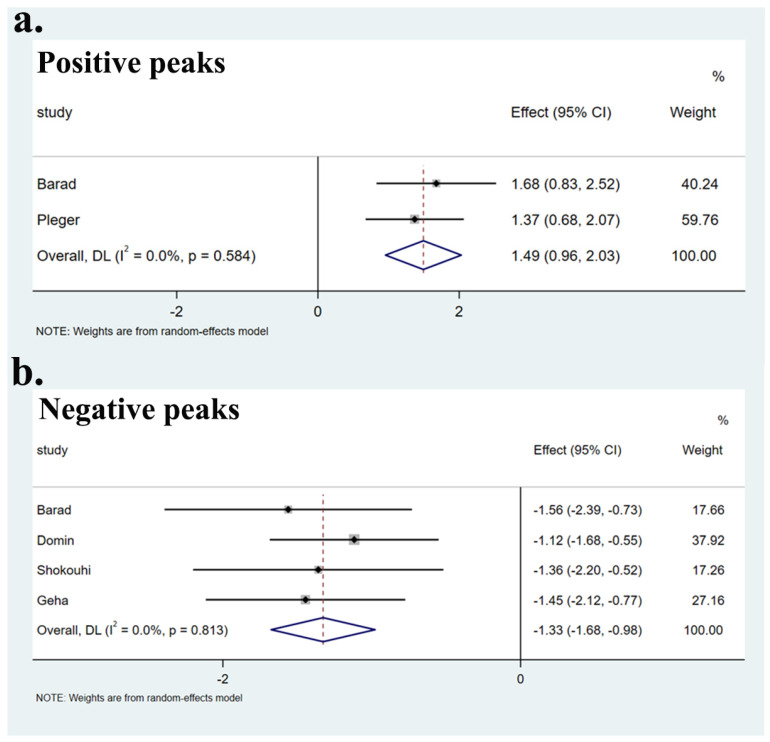
Forest plots of heterogeneity of main results. Heterogeneity assessment of the main results showed no heterogeneity in both positive peaks (**a**) and negative peaks (**b**) (*p* > 0.05).

**Figure 4 brainsci-12-01115-f004:**
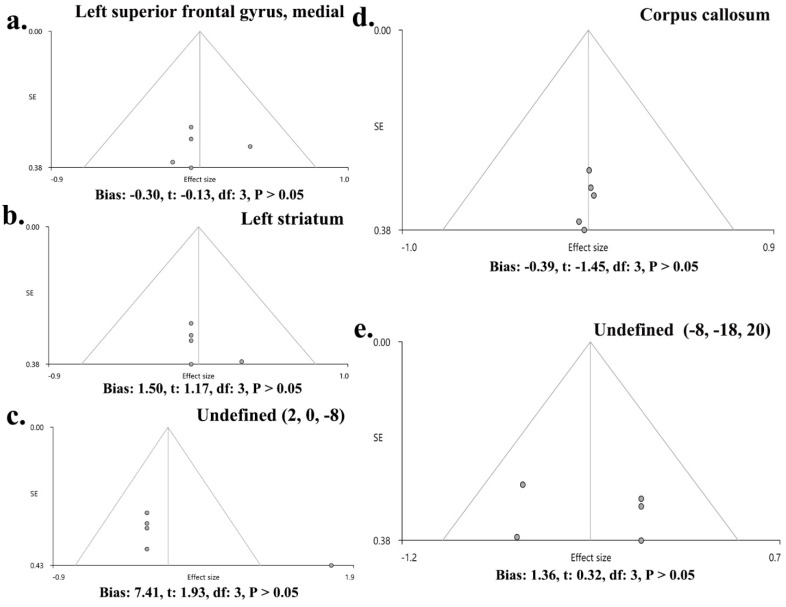
Funnel plots of publication bias in the main results. The main results showed no publication bias in SFGmedial (bias: −0.30, t: −0.13, df: 3, *p* > 0.05) (**a**), left striatum (bias: 1.50, t: 1.17, df: 3, *p* > 0.05) (**b**), undefined area (2, 0, −8) (bias: 7.41, t: 1.93, df: 3, *p* > 0.05) (**c**), CC (bias: −0.39, t: −1.45, df: 3, *p* > 0.05) (**d**), and undefined area (−8, −18, 20) (bias: 1.36, t: 0.32, df: 3, *p* > 0.05) (**e**).

**Figure 5 brainsci-12-01115-f005:**
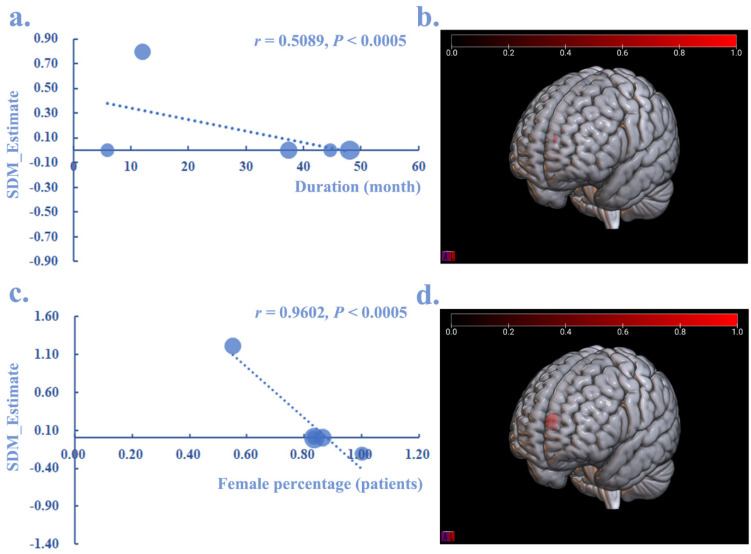
The meta-regression analysis between GMV alteration in CRPS and disease duration, as well as the percentage of female patients. Both disease duration (*r* = 0.5089, *p* < 0.0005) (**a**,**b**) and the percentage of female patients (*r* = 0.9602, *p* < 0.0005) (**c**,**d**) showed a negative relationship with the increased (red) GMV in SFGmedial.L.

**Table 1 brainsci-12-01115-t001:** Demographic information of five studies included.

Study	Year	Participants (Female)		Age, Year Means (SD)		Duration, Month Means (SD)	Pain Score (Means)	12-Point Checklist (Score)
		CRPS	HCs	CRPS	HCs			
Barad et al. [13]	2014	15 (15)	15 (15)	44.0 (NA)	44.1 (NA)	44.67 (13.96) *	VAS (7.25)	9.5
Domin et al. [14]	2021	24 (20)	33 (19)	50.75 (14)	54.42 (13.49)	48.12 (37)	VAS (4.8)	11.0
Pleger et al. [1]	2014	20 (11)	20 (11)	41.8 (9.8)	41.6 (9.6)	11.95 (3.20) *	NRS (5.3)	9.5
Shokouhi et al. [12]	2018	12 (10)	16 (10)	51.1 (12.7)	44.4 (11.6)	5.9 (2.9)	BPI (3.8)	10.5
Geha et al. [11]	2008	22 (19)	22 (19)	40.7 (2.3)	40.5 (2.3)	37.42 (7.90) *	VAS (5.76)	10.0

*: Calculated with the provided data of study included; VAS: Visual Analog Scale; NRS: Numeric Rating Scale; BPI: Brief Pain Inventory questionnaire; NA: not available.

**Table 2 brainsci-12-01115-t002:** Technical information of five studies included.

Study	Year	Method	Space	Magnetic Field (T)	Voxel Size (mm^3^)	Main Findings of GMV Abnormality in CRPS	*p*-Value
Barad et al. [13]	2014	VBM	MNI	3.0	1.5 × 1.5 × 1.5	Lesser: L orbitofrontal cortex, L mid-cingulate cortex, R mid-cingulate cortex, L posterior mid-cingulate cortex, L dorsal insula, L anterior mid-cingulate cortex Greater: R hypothalamus, L dorsal putamen, L inferior temporal lobe	*p*_FDR_ < 0.0005
Domin et al. [14]	2021	VBM	MNI	3.0	1 × 1 × 1	Lesser: bilateral thalamus	*p*_FWE_ < 0.05
Pleger et al. [1]	2014	VBM	MNI	1.5	1 × 1 × 1	Greater: dorsomedial prefrontal cortex	*p*_FWE_ < 0.05
Shokouhi et al. [12]	2018	VBM	MNI	3.0	1 × 1 × 1	Lesser: R supramarginal gyrus, R fusiform gyrus, R supplementary motor area	*p*_FWE_ < 0.05
Geha et al. [11]	2008	VBM	MNI	3.0	1 × 1 × 1	Lesser: insula	*p*_CORR_< 0.05

GMV: gray matter volume; VBM: voxel-based morphometry; MNI: Montreal Neurological Institute; FDR: false discovery rate; FWE: family-wise error; CORR: corrected.

**Table 3 brainsci-12-01115-t003:** Significant and consistent GMV abnormality in the main analysis.

Brain Areas	MNI Coordinate	SDM-Z	*p*-Value	Voxels	Local Peaks	Jackknife Analysis
**CRPS > HCs**						
Left superior frontal gyrus, medial	2, 52, 20	1.005	<0.001	324	Left superior frontal gyrus, medial Right anterior cingulate/paracingulate gyri, BA 32	3/5
Left striatum	−22, −4, −6	1.029	<0.001	79	Left striatum	4/5
Undefined	2, 0, −8	1.034	<0.001	56	Undefined (2, 0, −8) Undefined (2, −4, −10)	3/5
**CRPS < HCs**						
Corpus callosum	−2, −18, 26	−1.769	<0.001	1601	Corpus callosum (−2, −18, 26) Corpus callosum (−6, −12, 28) Corpus callosum (−2, −12, 28) Right supplementary motor area, BA 4 Right median cingulate/paracingulate gyri, BA 23 Right median cingulate/paracingulate gyri	4/5
Undefined	−8, −18, 20	−1.693	<0.001	312	Undefined (−8, −18, 20) Undefined (−8, −12, 20) Corpus callosum (−4, −20, 20) Corpus callosum (0, −18, 22) Undefined (−6, −8, 18) Undefined (0, −4, 20) Right caudate nucleus Undefined (12, −8, 18) Undefined (2, −12, 12) Right thalamus Right anterior thalamic projections	3/5

BA: Brodmann area.

**Table 4 brainsci-12-01115-t004:** Jackknife analysis of main analysis results.

Study	CRPS > HCs (GMV)			CRPS < HCs (GMV)	
	Left Superior Frontal Gyrus, Medial	Left Striatum	Undefined (2, 0, −8)	Corpus Callosum	Undefined (−8, −18, 20)
Barad	Yes	No	No	Yes	No
Domin	No	Yes	No	No	No
Geha	Yes	Yes	Yes	Yes	Yes
Pleger	No	Yes	Yes	Yes	Yes
Shokouhi	Yes	Yes	Yes	Yes	Yes
Total	3/5	4/5	3/5	4/5	3/5

## Data Availability

The data presented in this study are available on request from the corresponding author.

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
