# Peer review of "Gray Matter Abnormalities in Patients with Complex Regional Pain Syndrome: A Systematic Review and Meta-Analysis of Voxel-Based Morphometry Studies"

_brainsci, 2022, doi:10.3390/brainsci12081115_

Round 1
Reviewer 1 Report
Strengths:
-Clinically relevant and focused study question
-PRISMA checklist utilized
-Protocol was registered
-Comprehensive literature search (PubMed, Web of Science, and MEDLINE) and current (through 1/27/22)
-Heterogeneity, Sensitivity, and Publication Bias all assessed
Limitations:
-5 studies analyzed, limited external validity
Suggestions:
-Remove last paragraph under 2.5
-Include a statement in the Discussion regarding how the patient cohorts amongst the five studies compare to one another as far as sociodemographic factors, prior medications, prior interventions, prior behavioral health history/comorbidities (prevalence of anxiety/depression), and other potential confounding variables that may have affected the analysis etc. For differences uncovered, comment on them in the limitations section. For strengths found, emphasize how this may help strengthen the internal validity of the study findings and conclusions reached.
Author Response
Point to point responses
Thanks very much for the hard work of the reviewer. All questions were detailly answered. New manuscript has been revised and re-submitted. The point-to-point responses were described as follows:
Q1: 5 studies analyzed, limited external validity.
R1: This is indeed a limitation of our study. We completely agree that large datasets will be better for meta-analysis. Unfortunately, CRPS is a rare disease that is explored by small part of studies. Therefore, the small samples were included in this neuroimaging meta-analysis study. For the purpose of enhancing the robust of results, conservative threshold and jackknife/heterogeneity analysis were performed in our study. Taken together, main findings in this meta-analysis may provide a reference for readers, which were suggested to be accepted cautiously.
Q2: Remove last paragraph under 2.5.
R2: Thanks for your constructive suggestion. The last paragraph under 2.5 has been removed in the revised manuscript.
Q3: Include a statement in the Discussion regarding how the patient cohorts amongst the five studies compare to one another as far as sociodemographic factors, prior medications, prior interventions, prior behavioral health history/comorbidities (prevalence of anxiety/depression), and other potential confounding variables that may have affected the analysis etc. For differences uncovered, comment on them in the limitations section.
R3: Thanks for your constructive suggestion. We completely agree that these factors may play an effect on the GMV alterations involved in CRPS.
The statement has been included in the discussion, which was described as follows:
“It must also be pointed out that the sociodemographic factors, prior interventions (e.g., medication), prior behavioral health history/comorbidities (prevalence of anxiety/depression), and other potential confounding variables that were unavailable in included studies might also play an effect on the brain structural alterations in patients with CRPS.”
The shortage has been acknowledged in limitations:
“Thirdly, meta-regression results were suggested to be accepted cautiously because of lesser datasets, as well as the risk factors were suggested for comprehensive exploration with larger datasets”
Q4: For strengths found, emphasize how this may help strengthen the internal validity of the study findings and conclusions reached.
R4: Thanks for the suggestion. In order to enhance the reliability of results, we separately performed the Q test, jackknife sensitivity, Egger test to check the heterogeneity, robustness, and publication bias of main findings, which were commonly adopted in meta-analysis studies (Xu et al., 2020, Eur J Neurol; Lim et al., 2020, Neurosci Biobehav Rev). The effect powers were generally accepted (Liberati et al., 2009, BMJ). In addition, the purposes of these methods were respectively described in methods, which were showed as follows:
“The default Q maps of SDM were used to check the specific brain regions that present between-study heterogeneity when comparing with a global set of voxels. The P threshold, peak height Z and cluster extent in the random-effects model were set to 0.005, 1, and 10 voxels, respectively (21). The robustness of the meta-analysis results was assessed by a whole-brain voxel-based jackknife sensitivity analysis, which was performed by repeating the same steps as the main analysis after removing each study, one by one. Additionally, the total repeated times of the main results in all or most analysis results were regarded as reliable. Potential publication bias was checked by the Egger test of SDM default. The P value less than 0.05 was regarded as having publication bias.”
Reviewer 2 Report
Reviewer Comments on Manuscript Brainsci-1719055
General Comments: This manuscript conducts a meta-analysis of CRPS VBM studies. However, the number of studies which pass the authors criteria for inclusion is very small. The results are marginal and could easily be caused by defects in data analysis.
Specific Comments
1) The manuscript should be proof-read someone who is proficient in English. There are too many instances to list of improper/colloquial sentence construction. For example, see Section 2.2. The sentence “Both demographic and technical information were collected and showed respectively in this study” could be better written as “Both demographic and technical information were collected and presented”
2) Section 2.5: Remove the sentences “3. Results This section may be divided by subheadings. It should provide a concise and precise description of the experimental results, their interpretation, as well as the experimental conclusions that can be drawn.”
3) Section 3.1: The number of studies used for meta-analysis seems to be much lower than what is recommended for neuroimaging studies (see Muller et al in Neuroscience and Biobehavioral Reviews 84 (2018) 151–161. Can anything meaningful be obtained from just 5 studies?
4) Figure 2: Most of the areas of increased GMV seem too small to interpret. The center of mass of the undefined area with decreased GMV seems to be in the ventricles. The reductions in GM reported seem restricted to the inter-hemispheric fissure. Corpus Callosum is a white matter structure. There is no gray matter in it. This indicates that the FWHM of the meta-analysis may be too large, when combined the smoothness applied to estimates in the constituent studies
5) 5 replicates may not be sufficient for performing jackknife analysis, especially when is examining high-dimensional neuroimaging datasets
6) Figure 3: Since the number of studies is very small, the Cochran Q is not a reliable measure of heterogeneity (see Gavaghan, et al, Pain 2000;85:415-24.)
7) Figure 4: The publication bias estimate has only 3 degrees of freedom, which is woefully low for parametric statistics. What is the effect size/power of this estimate?
8) Figure 5: There is a large outlier in the plots. Without the outlier, there does not seem to be any significant effects of either sex-ratio or disease duration on the SDM-estimate.
Discussion
The results seem counter intuitive. How can increased gray matter volume result in brain pathology in CRPS? Compensation for functional deficits does not usually involve increased gray matter volumes.
Author Response
Point to point responses
Thanks very much for the hard work of the reviewer. All questions were detailly answered. New manuscript has been revised and re-submitted. The point-to-point responses were described as follows:
Q1: This manuscript conducts a meta-analysis of CRPS VBM studies. However, the number of studies which pass the authors criteria for inclusion is very small. The results are marginal and could easily be caused by defects in data analysis.
R1: This is indeed a limitation of our study. We completely agree that large datasets will be better for meta-analysis. Unfortunately, CRPS is a rare disease that is explored by a small part of studies. Therefore, only small sample sizes were included in this neuroimaging meta-analysis study. For the purpose of enhancing the robust of results, conservative threshold and jackknife/heterogeneity analysis were performed in our study. Taken together, the significant results in this meta-analysis may be a reference for readers, which were suggested to be accepted cautiously.
Q2: The manuscript should be proof-read someone who is proficient in English. There are too many instances to list of improper/colloquial sentence construction. For example, see Section 2.2. The sentence “Both demographic and technical information were collected and showed respectively in this study” could be better written as “Both demographic and technical information were collected and presented”
R2: Thanks for the suggestion of reviewer. We are so sorry for any inconvenience because of language question. The language of manuscript has been polished by the language editing services of MDPI. The new manuscript has been re-submitted.
Q3: Section 2.5: Remove the sentences “3. Results This section may be divided by subheadings. It should provide a concise and precise description of the experimental results, their interpretation, as well as the experimental conclusions that can be drawn.”
R3: Thanks for the suggestion. The last paragraph under 2.5 has been removed in the revised manuscript.
Q4: Section 3.1: The number of studies used for meta-analysis seems to be much lower than what is recommended for neuroimaging studies (see Muller et al in Neuroscience and Biobehavioral Reviews 84 (2018) 151–161. Can anything meaningful be obtained from just 5 studies?
R4: This is a constructive question. We completely agree that only 5 included studies may be less in the neuroimaging meta-analysis. Consistent with the previous meta-analysis with small sample (Zhou C., 2021, Front Endocrinol), we adopted more conservative threshold, as well as performed jackknife sensitivity and Q test heterogeneity assessment to enhance the reliability of findings in this study. We must acknowledge that this “reliability” is relative, and larger datasets will be pooled into the meta-analysis in the future. Until this rare disease is fully studied, our findings may provide a reference for readers.
Q5: Figure 2: Most of the areas of increased GMV seem too small to interpret. The center of mass of the undefined area with decreased GMV seems to be in the ventricles. The reductions in GM reported seem restricted to the inter-hemispheric fissure. Corpus Callosum is a white matter structure. There is no gray matter in it. This indicates that the FWHM of the meta-analysis may be too large, when combined the smoothness applied to estimates in the constituent studies.
R5: Thanks for your constructive suggestion. The FWHM was adopted with the SDM default parameter that has been balanced for the most ideal (Radua J., 2012, European Psychiatry; Radua J., 2014, Frontiers in Psychiatry). The reason why peaks located in white matter area is that the SDM algorithm does not restrict its calculations to voxels within a mask (e.g., gray matter), which was detailly explanted in the previous study (Radua J., 2014, Frontiers in Psychiatry). Since a few studies were included in this meta-analysis, we try to critically and comprehensively report the peak and sub-peak coordinates to make readers understand the findings better.
Q6: 5 replicates may not be sufficient for performing jackknife analysis, especially when is examining high-dimensional neuroimaging datasets
R6: Thanks for your suggestion. Jackknife sensitivity analysis was used to check the reliability of results in this meta-analysis by iteratively removing the included study. Since 5 studies were included in this study, the replicated times for the same meta-analysis was limited to a maximum of 5. Therefore, the main results were consistent in all or most repeated analysis may be regarded as robust (Radua et al., 2009, Br J Psychiatry). But we must admit that only 5 studies included was a limitation for robust findings.
Q7: Figure 3: Since the number of studies is very small, the Cochran Q is not a reliable measure of heterogeneity (see Gavaghan, et al, Pain 2000;85:415-24.)
R7: This is a constructive question. According to the previous explanation about the SDM software: “It must be also commented that regions with differences between patients and controls may be falsely detected as heterogeneous because of the discrepancy between the real effect sizes from studies reporting peaks and the null effect sizes from studies not reporting peaks.” (Radua J., 2012, European Psychiatry), the findings in SDM meta-analysis were more likely regarded as heterogeneity. Therefore, Q test in our study showed no heterogeneity in our positive and negative peaks can be reliable.
Q8: Figure 4: The publication bias estimate has only 3 degrees of freedom, which is woefully low for parametric statistics. What is the effect size/power of this estimate?
R8: This is a constructive question. We must admit that the power is limited because of small sample. As we all known, heterogeneity assessment and sensitivity analysis were commonly performed in meta-analysis to enhance the reliability of findings (Liberati et al., 2009, BMJ). In order to provide a more reliable reference where possible, we performed the Egger test to check the publication bias of main results. The same way of checking the publication bias has been reported in previous neuroimaging meta-analysis study with small sample (Yang et al., 2021, Front Neurol). In addition, we also acknowledged the limitation in limitation: “fifthly, the lesser studies included attenuated the reliability of main findings in this study.”
After establishing the masks of main results, the SDM default algorithm then extracted the masks information and generated the SDM _ Estimate, which was the effect size in funnel plot.
Q9: Figure 5: There is a large outlier in the plots. Without the outlier, there does not seem to be any significant effects of either sex-ratio or disease duration on the SDM-estimate.
R9: Thanks for the constructive question. It is worth noting that meta-regression results were suggested to be accepted cautiously. We agree that lesser included studies may cause the large outlier that may play an effect on the findings. Thus, we acknowledge this shortage in the limitation and give a suggestion for cautious acceptance. In addition, we critically discussed the meta-regression results by combing the clinical evidence in the discussion.
Discussion “Interestingly, our meta-analysis results found higher GMV in the SFGmedial.L in CRPS patients than HCs, which demonstrated a negative relationship with the disease duration and percentage of female patients…Combined with the evidences that long-term chronic pain stimulation can also induce a series of emotional disorders (e.g., anxiety, depression) (30, 31) and the female predominance in CRPS, we speculated that the increased GMV in SFGmedial may be a functional compensation in patients with CRPS, which may be attenuated by the disease duration and female ratio.”
Limitation “Thirdly, meta-regression results should be treated cautiously due to the small size of the datasets”
Q10: The results seem counter intuitive. How can increased gray matter volume result in brain pathology in CRPS? Compensation for functional deficits does not usually involve increased gray matter volumes.
R10: This is an interesting question. In order to explanate the underlying mechanism of increased GMV within CRPS, animal study may be necessary by establishing the pain model and performing the inter-group comparison, which may not be accomplished by meta-analysis. However, this suggestion indeed inspires our further exploration of the potential mechanism of increased GMV involved in various pathological statements including CRPS. Furthermore, the speculation of functional compensation in specific brain region was based on the clinical evidences, main analysis results, as well as additional analysis in our study to provide a rational explanation for CRPS-related brain structural alterations. The possible compensation of increased GMV in specific brain regions has been described in previous study (Qiu et al., 2019, Hum Brain Mapp).
Reviewer 3 Report
Please see attached document.

Author Response
Point to point responses
Thanks very much for the hard work of the reviewer. All questions were detailly answered. New manuscript has been revised and re-submitted. The point-to-point responses were described as follows:
Q1: I believe however, that a major revision of the current report is needed for it to be suited for publication. It is my belief that the manuscript would benefit from a more detailed and physiologically oriented background and discussion, other than an enumeration of VBM changes. Also, the results should be more carefully interpreter and authors should be more precautious with their conclusions. I also have concerns regarding the written form of the manuscript. The English is not clear, and grammar is wrongly used many times throughout the manuscript. I believe for it to be fit for publication a major language review needs to take place.
R1: Thanks for your constructive suggestions.
Introduction has been revised as follows:
“Neuroimaging studies have focused on the brain structure alterations involved in CRPS. Voxel-based morphometry (VBM) is one of the most important methods for measuring gray matter volume (GMV) abnormalities (9, 10). According to the accumulating VBM evidence, patients with CRPS showed the GMV abnormalities in specific brain regions (e.g., cingulate cortex and amygdala) that played the key role in somatosensory and emotional functions (7). In addition, neuroimaging study reported the GM atrophy might be limited in the emotion-related brain regions including insula, ventromedial prefrontal cortex (VMPFC), and nucleus accumbens (NAc) (11). In the early stage of CRPS, patients showed the decreased GMV in cerebral areas that involved in the spatial body perception, somatosensory, and limbic system (12). Although brain structure abnormalities have been found to be closely related to the dysfunctions within CRPS, the findings of GMV alterations in specific brain regions can be heterogenous and controversial. For instance, it was reported that GMV of dorsomedial prefrontal cortex in patients with CRPS was higher than that of HCs (1). Barad et al. found that patients with CRPS had higher GMV in right hypothalamus, left dorsal putamen, and left inferior temporal lobe, as well as lower GMV in left orbitofrontal cortex, left middle cingulate cortex, right middle cingulate cortex, left posterior middle cingulate cortex, left dorsal insula, and left anterior middle cingulate cortex at the same time (13). Due to the limited research samples, heterogeneous and unreliable findings may be existed in CRPS studies (14). Therefore, it is of great necessity to perform a systematic review and meta-analysis to explore the reliable GMV alterations in CRPS.”
Discussion has been revised as follows:
“The SFGmedial is associated with various neuropsychological disorders, including schizophrenia (SCZ), obsessive compulsive disorder (OCD), attention deficit hyperactivity disorder (ADHD), and bipolar disorder (27-29). Combined with the evidences that the long-term chronic pain stimulation can also induce a series of emotional disorders (e.g., anxiety, depression) (30, 31) and the female predominance in CRPS, we speculated that the increased GMV of SFGmedial might be a functional compensation in patients with CRPS, which may be attenuated by the disease duration and female ratio.”
“According to the neuroimaging evidences, we found the undefined area (2, 0, -8) may be in the hypothalamic area (35, 36). Interestingly, the clinical and neuroimaging evidences showed the altered neurotransmitter system involved in hypothalamus and functional connectivity alterations between hypothalamus and distal nuclei may be an incentive for pain-related aversions (e.g., fatigue, and emotional alterations), as well as a biomarker for migraine occurrence (37). Therefore, the brain structural alteration within hypothalamus may be a potential mechanism for CRPS-related clinical or neurophysiological symptoms.”
“In our study, the GMV of the SMA.R and THA.R was significantly decreased, which was consistent with the previous meta- analyses of neuropathic pain (38, 39). Krainik et al. identified the important role of SMA in motor development in patients undergoing medial frontal lobectomy (40). Motor skills impairment has also been found in healthy adults receiving SMA.R-guided transcranial magnetic stimulation (nTMS) (41). Taken together, CRPS may induce the motor dysfunction of patients by impairing the brain structure of SMA.R. Interestingly, patients with hypoactivation of SMA showed the same motor dysfunction as medial frontal lobectomy, which suggested the possible functional compensation of greater GMV in the SFGmedial to withstand motor dysfunction (40).”
We are so sorry for any inconvenience because of language question. The language of manuscript has been polished by the language editing services of MDPI. The new manuscript has been re-submitted.
Q2: Abstract, Line 38: were significantly involved in the process of CRPS. I suggest the authors refrain from such conclusion. The identified changes in gray matter associate/ are present in CRPS patients, however its involvement in the process of the disease is not evaluable by cross-sectional data.
R2: Thanks for your constructive suggestion. We have revised the conclusion as:
“Brain structure abnormalities in sensorimotor regions (e.g., SFGmedial.L, SMA.R, CAU.R, MCC.R, and THA.R) may be susceptible in patients with CRPS.”
Q3: Introduction, line 47: One of the most important impairments in CRPS is pain-induced neuroplasticity in central nervous system (CNS). Neuropsychological symptoms are commonly found in CRPS patients without brain injury (5). Brain function alteration has been widely reported in clinical (6, 7). Though the rational of the authors is understandable this information is not clinically or physiologically precise, and more detail should be given when using such background.
R3: Thanks for your constructive suggestion. The text has been revised as:
“One of the significant impairments of CRPS is pain-induced neuroplasticity in the central nervous system (CNS). For instance, it was reported that cognitive function impairments and abnormal body representation were showed in CRPS patients without brain injury (5). Resting-state functional magnetic resonance imaging (Rs-fMRI) studies have shown that CRPS involves significant default mode network (DMN) alterations by comparing CRPS and non-CRPS (6, 7).”
Q4: Introduction, from line 57-71: Authors enumerate all VBM changes published in CRPS, however they do not introduce the prospect of such changes – both in terms of neuroanatomic plausibility of– eg. VBM changes may translate histopathological changes “VBM-detected alterations in gray matter can be used as a surrogate marker for hippocampal damage” Suzuki et al, 2013 (DOI: 10.1016/j.neuroimage.2013.03.042); “Also, a brief introduction on the biological importance of such brain areas would be of interest for the reader.
R4: Thanks for your constructive suggestion. We have revised the Introduction by combing the biological importance and GMV alterations, which was described as follows:
“Neuroimaging studies have focused on the brain structure alterations involved in CRPS. Voxel-based morphometry (VBM) is one of the most important methods for measuring gray matter volume (GMV) abnormalities (9, 10). According to the accumulating VBM evidence, patients with CRPS showed the GMV abnormalities in specific brain regions (e.g., cingulate cortex and amygdala) that played the key role in somatosensory and emotional functions (7). In addition, neuroimaging study reported the GM atrophy might be limited in the emotion-related brain regions including insula, ventromedial prefrontal cortex (VMPFC), and nucleus accumbens (NAc) (11). In the early stage of CRPS, patients showed the decreased GMV in cerebral areas involved in the spatial body perception, somatosensory, and limbic system (12).”
Q5: Discussion, line 174: Authors present an “undefined area (2, 0, -8) was showed in CRPS patients”; though not identified through the present atlas used, an effort should be made to enlighten the reader using a distinct atlas or visual description of such location (as done latter in the same sentence for a distinct area “and undefined area (extending to CAU.R, and THA.R)”.
R5: Thanks for your constructive suggestion. According to the neuroanatomical evidence (Simon et al., 2020, J Clin Invest; Billot et al., 2020, Neuroimage), the description of undefined area (2, 0, -8) has been revised as undefined area (2, 0, -8) within hypothalamus and discussed in discussion. The details were showed as follows:
“According to the neuroimaging evidences, we found the undefined area (2, 0, -8) may be in the hypothalamic area (35, 36). Interestingly, the clinical and neuroimaging evidences showed the altered neurotransmitter system involved in hypothalamus and functional connectivity alterations between hypothalamus and distal nuclei may be an incentive for pain-related aversions (e.g., fatigue, and emotional alterations), as well as a biomarker for Migraine occurrence (37). Therefore, the brain structural alteration within hypothalamus may be a potential mechanism for CRPS-related clinical or neurophysiological symptoms.”
Q6: Discussion, line 209: “Combined with the evidence that the long-term chronic pain stimulation can also induce a series of emotional disorders (e.g., anxiety, depression) (30, 31) and the female predominance in CRPS, we can assume that the higher GMV in SFGmedial.L performs a functional compensation to resist the neuropsychological deficits caused by CRPS”. Neuropsychological deficits are not assessed in the current study; the direction of this adaptation/mal adaptation in CRPS patients cannot be inferred from this data. The authors may discuss the possible meaning of such change as a functional compensation but the possibility of this translating a disfunction or even being a pre-crps finding that may be a predisposition factor should be acknowledge. A more careful discussion regarding the meaning of this and other structural changes described should be attempted.
R6: Thanks for your suggestions. A more cautious discussion has been revised, which was showed as follows:
“Combined with the evidences that the long-term chronic pain stimulation can also induce a series of emotional disorders (e.g., anxiety, depression) (30, 31) and the female predominance in CRPS, we speculated that the increased GMV of SFGmedial might be a functional compensation in patients with CRPS, which may be attenuated by the disease duration and female ratio.”
Q7: Conclusion, line 242: “In conclusion, GMV changes of sensorimotor cerebral areas are the specific brain injury of CRPS”. This is an incorrect affirmation given the presented data – GMV changes in sensorimotor areas are present in the evaluated data sample, however these are present in many other conditions, thus not specific of CRPS, moreover, once again, causality cannot be affeered from this
data sample.
R7: Thanks very much for your constructive suggestion. We agree that this affirmation may be inappropriate, especially from a lesser sample. The conclusion has been revised as:
“In conclusion, the GMV impairments of sensorimotor cerebral areas may be susceptible in patients with CRPS.”
Q8: Statistical approach – Authors model GMV changes and percentages of female patients. The goal is to understand if sex has a relationship with GMV changes in CRPS. Modeling the percentage of female patients poses some questions – not only gender is a categorical variable, thus the percentage of patients is not a precise indicator, but also linearly modeling percentages may be problematic, given the superior bound seems relevant (many percentages >80%). The authors should acknowledge that the variance of a proportion (0-1) may have a variance that is usually not constant across the range of the dependent variable.
The ideal would be using a logistic regression; however, I understand that may not be possible given the data access limitation. Thus, if no other analytical option is available, I believe this should be carefully discussed as a limitation.
Still regarding the metanalytical regression – authors only present the results for the SFGmedial.L region. I believe the other regions were also tested. Authors should 1) correct for multiple comparisons given the number of linear regressions tested; 2) acknowledge the negative results for the other regions.
R8: Thank you very much for your kind remainder. It is true as the Reviewer suggest that the lineal model may not be the most ideal method for exploring the potential relationship between sexual difference and GMV alterations. However, the linear model meta-regression analysis was based on the SDM default algorithm that was used in previous meta-analysis VBM studies (Wan et al., 2021, Transl Psychiatry). Therefore, we cannot change the default of SDM, which is indeed a limitation.
By performing the meta-regression analysis, we indeed found some other brain regions affected by the risk factors possible. In order to provide the reliable findings for readers, we showed the brain regions that were significant in main analysis, this method which has been described in previous studies (Zhou et al., 2021, Front Endocrinol; Wang et al., 2021, J Affect Disord).
Q9: Overall, throughout the manuscript English needs revision; some of its grammar errors:
Title would benefit from correction to plural: Gray Matter Abnormalities (…) voxel-based morphometry studies.
Line 25, Abstract: significant gray matter volume (GMV) abnormalities. Plural for abnormalities should be used throughout the manuscript.
Line 44: Complex regional pain syndrome (CRPS) is a rare chronic pain that characterized with spontaneous and evoked regional pain in extremities, which is commonly caused by limb trauma. Should read: CRPS is a chronic pain syndrome that is characterized by (…).
Line 103: Any controversial things were consulted and solved with the third reviewer (LFY). Should read: Any controversial assessments were consulted and solved…
R9: Thanks for your suggestions. We are so sorry for any inconvenience because of the language question. All questions above have been polished in revised manuscript. The language of manuscript has been polished by the language editing services of MDPI. The new manuscript has been re-submitted.
Title has been revised as: “Gray Matter Abnormalities in Patients with Complex Regional Pain Syndrome: A Systematic Review and Meta-Analysis of Voxel-Based Morphometry Studies”
Line 25, Abstract has been revised as: “This study aimed to perform a systematic review and meta-analysis to explore the significant gray matter volume (GMV) abnormalities between patients with CRPS and healthy controls (HCs).”
Line 44 has been revised as: “Complex regional pain syndrome (CRPS) commonly caused by limb trauma is a chronic pain that is characterized by spontaneous and evoked regional pain in the extremities (1, 2).”
Line 103 has been revised as: “All work was independently accomplished by two reviewers (TM and ZYL), any controversial assessments were consulted and solved with the third reviewer (LFY).”
Round 2
Reviewer 3 Report
I believe the manuscript was amended to match my concerns. I agree with its publication in the current form.
Thank you